# Deferiprone (DFP) Targets Cancer Stem Cell (CSC) Propagation by Inhibiting Mitochondrial Metabolism and Inducing ROS Production

**DOI:** 10.3390/cells9061529

**Published:** 2020-06-23

**Authors:** Marco Fiorillo, Fanni Tóth, Matteo Brindisi, Federica Sotgia, Michael P. Lisanti

**Affiliations:** 1Translational Medicine, School of Science, Engineering and the Environment (SEE), University of Salford, Greater Manchester M5 4WT, UK; m.fiorillo@salford.ac.uk (M.F.); F.Toth@edu.salford.ac.uk (F.T.); 2Department of Pharmacy, Health and Nutritional Sciences, University of Calabria, Via P. Bucci, 87036 Rende (CS), Italy; matteo_brindisi@libero.it

**Keywords:** iron chelators, CSCs, ROS, deferiprone, mitochondrial metabolism, FDA-approved drugs

## Abstract

Deferiprone (DFP), also known as Ferriprox, is an FDA-approved, orally active, iron chelator that is currently used clinically for the treatment of iron-overload, especially in thalassaemia major. As iron is a critical factor in Fe-S cluster assembly that is absolutely required for the metabolic function of mitochondria, we hypothesized that DFP treatment could be used to selectively target mitochondria in cancer stem cells (CSCs). For this purpose, we used two ER(+) human breast cancer cell lines, namely MCF7 and T47D cells, as model systems. More specifically, a 3D tumorsphere assay was employed as a functional readout of CSC activity which measures anchorage-independent growth under low attachment conditions. Here, we show that DFP dose dependently inhibited the propagation of CSCs, with an IC-50 of ~100 nM for MCF7 and an IC-50 of ~0.5 to 1 μM for T47D cells, making DFP one the most potent FDA-approved drugs that we and others have thus far identified for targeting CSCs. Mechanistically, we show that high concentrations of DFP metabolically targeted both mitochondrial oxygen consumption (OCR) and glycolysis (extracellular acidification rates (ECAR)) in MCF7 and T47D cell monolayers. Most importantly, we demonstrate that DFP also induced a generalized increase in reactive oxygen species (ROS) and mitochondrial superoxide production, and its effects reverted in the presence of N-acetyl-cysteine (NAC). Therefore, we propose that DFP is a new candidate therapeutic for drug repurposing and for Phase II clinical trials aimed at eradicating CSCs.

## 1. Introduction

Cancer stem-like cells (CSCs) are a small sub-population of cancer cells that are thought to be responsible for driving treatment failure and poor clinical outcomes in nearly all cancer patients [1,2]. CSCs functionally behave as tumor-initiating cells (TICs) and are involved in the early phases of carcinogenesis. In addition, because of their ability to undergo anchorage-independent growth, they have also been implicated in the metastatic dissemination of aggressive tumor cells to distant sites [3]. Ultimately, the overall mortality in >90% of cancer patients has been attributed to distant metastasis, identifying CSCs as an important therapeutic target for life extension and survival [4,5]. Therefore, there is an unmet clinical need to identify new therapeutic approaches for the eradication of CSCs [6,7].

One of the key features that appears to be conserved among CSCs derived from different organ sites is their critical reliance on mitochondrial energy production, especially related to their 3D growth in an anchorage-independent fashion [8]. Importantly, anchorage-independent propagation is a required step in metastatic dissemination [9,10]. Therefore, the pharmacological inhibition of anchorage-independent growth may be a new Achilles’ heel for the prevention of the onset of metastatic disease in cancer patients [11]. As iron is a critical element that is required for the redox reactions that drive mitochondrial ATP production by oxidative phosphorylation (OXPHOS), we reasoned that FDA-approved iron chelators might represent a new therapeutic approach [12,13] to target mitochondria in CSCs [14].

Iron is found in the prosthetic groups of proteins, including iron-sulfur clusters and heme. Moreover, the iron within the heme portions of cytochromes b/c and cytochrome P450 are required for OXPHOS and the detoxification of chemicals, respectively [15,16]. Numerous enzymes contain iron-sulfur clusters, including mitochondrial Complex I and II, which are crucial for redox reactions involved in respiration and in ATP production [17]. Iron also plays a key role in reactions that lead to the generation of reactive oxygen species (ROS) [18]. The existence of di- and tri-valent states of iron permits iron to gain and lose electrons, which are essential steps that are required for electron transport. Donating electrons to oxygen is potentially one of the first causes of cellular toxicity induced by ROS formation [18,19].

Neoplastic cells have a higher requirement for iron than normal cells, in part due to their rapid rate of DNA synthesis, and this is highlighted by the increased expression of several iron-binding and transport proteins in cancer cells. For this reason, many in vitro and in vivo studies have demonstrated that iron chelators can inhibit tumor growth [13,20]. Traditionally, pro-oxidant-based agents that can increase ROS production or decrease antioxidant capacity in cancer cells have shown therapeutic potential in pre-clinical studies [21,22,23]. For example, cisplatin is largely used as a chemotherapeutic agent for cancer therapy because it induces ROS generation [24]. In comparison with normal stem cells and “bulk” cancer cells, very little is known regarding the levels of ROS in CSCs. The increased expression of free radical scavenger systems may be used by CSCs to lower intracellular ROS levels. Oxidative stress plays a key role in cancer cells, and exogenous agents may further increase ROS effectively and selectively kill CSCs [20].

Here, we evaluated the efficacy of 3-Hydroxy-1,2-dimethyl-4(1H)-pyridone, called Deferiprone (DFP), a known FDA-approved iron chelator [25,26], for the targeting of anchorage-independent growth, using MCF7 and T47D breast cancer cells as a model system. Importantly, we show that DFP potently inhibits 3D tumorsphere formation, with an IC-50 of ~100 nM (for MCF7 cells) and an IC-50 of ~500 nM (for T47D cells), by elevating ROS and mitochondrial superoxide production. However, DFP was not toxic in the total MCF7 and T47D cancer cell population, normal human fibroblasts (hTERT-BJ1), or the non-tumorigenic epithelial cell line (MCF10A) until levels of ~100 μM were reached. Therefore, it appears that CSCs are nearly 1,000 times more sensitive to the effects of DFP.

## 2. Materials and Methods

### 2.1. Cell Models and Other Reagents

Human breast cancer cell lines, MCF7 [ER(+)], T47D [ER(+)], and MCF10A, were obtained commercially from the American Type Culture Collection (ATCC; Manassas, VA, USA), while human immortalized fibroblasts (hTERT-BJ1) were originally purchased from Clontech, Inc (now Takara Bio USA, Inc., Mountain View, CA, USA). The MCF7 and hTERT-BJ1 cell lines were maintained in Dulbecco’s Modified Eagle Medium (DMEM; GIBCO, Paisley, UK) supplemented with 10% FBS, 1% Glutamax, and 1% Penicillin-Streptomycin. The T47D cell line was maintained in Dulbecco’s Modified Eagle Medium F12 (DMEM/F12; GIBCO) supplemented with 10% FBS and 1% Penicillin-Streptomycin. The MCF10A cell line was maintained in a mammary epithelial cell growth medium (MEGM; Lonza, Basel, Switzerland) supplemented with 0.4% Bovine pituitary extract (BPE), 0.1% insulin, 0.1% hEGF, 0.1% Hydrocortisone, 0.1% GA-1000, and 100 ng/mL of cholera toxin. All the cell lines were maintained at 37 °C in 5% CO_2_. The 3-Hydroxy-1,2-dimethyl-4(1H)-pyridone (DFP) was purchased from Sigma-Aldrich, Inc.

### 2.2. Cell Viability

Cell viability was assessed by a sulphorhodamine (SRB) assay based on the measurement of cellular protein content. After treatment with DFP for 5 days in 96-well plates, cells were fixed with 10% trichloroacetic acid (TCA) for 1 h in a cold room and dried overnight at room temperature. Then, the cells were incubated with SRB for 15 min, washed twice with 1% acetic acid, and air dried for at least 1 h [1]. Finally, the protein-bound dye was dissolved in 10 mM of Tris pH 8.8 solution and read using a plate reader at 540 nm.

### 2.3. Mammosphere Formation Efficiency (MFE)

A single cell suspension was prepared using enzymatic (1x Trypsin-EDTA, Sigma Aldrich, #T3924) and manual disaggregation (25 gauge needle) to create a single-cell suspension [2]. The cells were plated at a density of 500 cells/cm^2^ in mammosphere medium (DMEM-F12/B27/20ng/mL EGF/PenStrep) in non-adherent conditions in culture dishes coated with (2-hydroxyethylmethacrylate) (poly-HEMA, Sigma, #P3932). Then, the cells were treated with increasing concentrations of DFP (in the range 10 nM to 10 μM). Vehicle alone (DMSO) control cells were processed in parallel. We also tested the concentrations of DFP (1 μM and 10 μM) in the presence of 1 mM and 5 mM of *N*-acetyl-cysteine (NAC). The cells were grown for 5 days and maintained in a humidified incubator at 37 °C. After 5 days for culture, spheres >50 μM were counted using an eye piece graticule, and the percentage of cells plated which formed spheres was calculated; the percentage is referred to as the percentage mammosphere formation and was normalized to one (1 = 100% MFE) [3]. Similar results were also obtained when the cells were seeded at a density of 200 cells/cm^2^.

### 2.4. ALDEFLUOR Assay

The level of ALDH activity was assessed by using the fluorescent reagent ALDEFLUOR. The ALDEFLUOR kit (StemCell technologies, Durham, NC, USA) was used to detect the cell sub-populations with various amounts of ALDH enzymatic activity by flow cytometry (SONY SH800). Briefly, 1 × 10^5^ cells were incubated in 1 mL of ALDEFLUOR assay buffer containing ALDH substrate (5 μL/mL) for 40 min at 37 °C. In each experiment, a sample of cells was stained under identical conditions, with 30 μM of diethylaminobenzaldehyde (DEAB), a specific ALDH inhibitor, as a negative control. The ALDH-positive population was established according to the manufacturer’s instructions and evaluated using 50,000 cells. All the ALDH experiments were performed three times independently.

### 2.5. Seahorse XFe-96 Metabolic Flux Analysis

The extracellular acidification rates (ECAR) and real-time oxygen consumption rates (OCR) for cells treated with DFP were assessed using the Seahorse Extracellular Flux (XFe-96) analyzer (Seahorse Bioscience, MA, USA). A total of 10,000 cells (for 24 h and 48 h time points) and 5000 cells (for 72 h and 120 h time points) per well were seeded into XFe 96-well cell culture plates and incubated overnight to allow attachment. The cells were then treated with DFP (from 10 μM to 1000 μM) for 24 to 120 h. The vehicle alone (DMSO) control cells were processed in parallel [4]. After time treatment, the cells were washed in pre-warmed XF assay media (or, for the OCR measurement, XF assay media supplemented with 10 mM of glucose, 1 mM of Pyruvate, ad 2 mM of L-glutamine and adjusted at 7.4 pH). The cells were then maintained in 175 μL/well of XF assay media at 37 °C in a non-CO_2_ incubator for 1 h. During the incubation time, we loaded 25 μL of 80 mM glucose, 9 μM of oligomycin, and 1 M of 2-deoxyglucose (for ECAR measurement) or 10 μM of oligomycin, 9 μM of FCCP, 10 μM of rotenone, and 10 μM of antimycin A (for OCR measurement) in XF assay media into the injection ports in the XFe-96 sensor cartridge. The measurements were normalized by protein content (SRB assay) [5]. The dataset was analyzed by XFe-96 software and GraphPad Prism software using a one-way ANOVA and Student’s t-test calculations. All the experiments were performed in quintuplicate, three times independently. The energy plots (cell energy phenotype test) shown in Figure 5c and Figure 6c, were generated by following the manufacturer’s guidelines and instructions (Seahorse, Agilent Technologies).

### 2.6. ROS Staining

The reactive oxygen species (ROS) production was measured using CM-H_2_DCFDA (C6827, ThermoFisher, Waltham, USA), a cell-permeable probe that is non-fluorescent until oxidation within the cell, and a Total Reactive Oxygen Species (ROS) Assay Kit 520 nm (ThermoFisher). The MCF7 cells were treated with DFP (1 μM to 1000 μM) for 24 to 120 h. The vehicle alone (DMSO) control cells were processed in parallel. After 48 h, the cells were washed with PBS and incubated with CM-H_2_DCFDA (diluted in PBS/CM to a final concentration of 1 μM) for 20 min at 37 °C. The same procedure was performed using the Total Reactive Oxygen Species (ROS) Assay Kit 520 nm (ThermoFisher), with an incubation time of 1 h at 37 °C. All subsequent steps were performed in the dark [6]. The cells were rinsed, harvested, and re-suspended in PBS/CM. The cells were then analyzed by flow cytometry using the SH800 (SONY). The ROS levels were estimated by using the mean fluorescent intensity of the viable cell population. The results were analyzed using the FlowJo software (BD Bioscience, San Jose, CA, USA).

### 2.7. Mitochondrial Superoxide Assessment

To evaluate the mitochondrial superoxide production, a MitoSOX (ThermoFisher) probe was used. The MCF7 cells were treated with DFP (1 μM to 1000 μM) for 24 to 120 h. The vehicle alone (DMSO) control cells were processed in parallel. After 48 h, the cells were washed with PBS and incubated with MitoSOX (diluted in PBS/CM to a final concentration of 5 μM) for 10 min at 37 °C. All subsequent steps were performed in the dark. The cells were rinsed, harvested, and re-suspended in PBS/CM. The cells were then analyzed by flow cytometry using the SH800 (SONY). The mitochondrial superoxide levels were estimated by using the mean fluorescent intensity of the viable cell population. The results were analyzed using the FlowJo software (BD Bioscience).

### 2.8. Statistical Analysis

All the analyses were performed with GraphPad Prism 7. The data were represented as mean ± SD (or ± SEM where indicated). All the experiments were conducted at least 3 times independently, with >3 technical replicates for each experimental condition tested (unless stated otherwise—e.g., when representative data is shown). Statistically significant differences were determined using the Student’s *t*-test or the analysis of variance (ANOVA) test. For the comparison among multiple groups, one-way ANOVAs were used to determine the statistical significance. *p* ≤ 0.05 was considered significant and all the statistical tests were two-sided.

## 3. Results

### 3.1. Evaluating the Effects of DFP on Cell Survival

To evaluate the effects of DFP on the cell viability/survival, we used the SRB assay to measure the protein content. As cells detach after undergoing apoptosis, this provides a sensitive assay for quantitating the relative amount of cells that remain attached to the cell culture plates. Figure 1 shows that DFP dose dependently inhibited the cell viability in the MCF7 and T47D cell monolayers after 5 days of treatment, with an IC-50 between 75 and 100 μM. In contrast, ~70% of the hTERT-BJ1 fibroblasts and ~100% of the MCF10A remained viable at 100 μM, while only 35% of MCF7 and ~50% of T47D remained viable at this concentration. Thus, DFP showed a preferential selectivity for targeting cancer cells.

### 3.2. Effects of DFP on CSC Propagation and ALDH Activity

We next used the 3D tumorsphere assay to as a read-out for CSC activity. This assay measures the functional ability of CSCs to undergo anchorage-independent growth under low-attachment conditions, which is a critical step that is mechanistically required for metastatic dissemination [8,9,10,11,12,13,14,15,16,17,18,19,20,21,22,23,24,25,26,27,28]. Figure 2A shows that DFP inhibits anchorage-independent growth remarkably well, with an IC-50 of ~100 nM for MCF7 cells and an IC-50 of ~500 nM for T47D cells after 5 days of treatment. Therefore, we can estimate that CSCs are approximately 1000-fold more sensitive to DFP than the “bulk” cancer cell population. In addition, we evaluated the CSCs’ formation in the presence of NAC. Interestingly, we observed that the DFP-induced reduction in the 3D tumorsphere formation reverted in the presence of 1 mM and 5 mM of NAC (Figure 2). Additionally, we used the ALDH activity to further validate the effects of DFP on CSCs [29]. Figure 3b demonstrates that 50 μM of DFP reduced the ALDH activity by >75% after 5 days of treatment. As ALDH is a metabolic marker of Epithelial-Mesenchymal Transition (EMT), this provides additional supporting evidence that DFP indeed targets the “stemness” phenotype of CSCs.

### 3.3. Effects of DFP on Mitochondrial Metabolism and Glycolysis

To quantitatively determine the effects of DFP on cell metabolism, we next used the Seahorse XFe96 to directly measure oxygen consumption rates (OCR) and extracellular cell acidification rates (ECAR) in MCF7 and T47D cell monolayers [30,31]. OCR is a surrogate marker for mitochondrial OXPHOS and ATP production, while ECAR is a measure of glycolytic flux. Figure 4 illustrates that DFP dose dependently inhibited mitochondrial metabolism in MCF7 cells at concentrations from 10 μM to 1 mM and slightly affected the glycolytic metabolism at 1 mM over a time period of one to three days. Figure 5 shows that similar results were obtained after 5 days of prolonged treatment. Note that 100 μM of DFP was sufficient to nearly completely inhibit the basal and maximal respiration, as well as ATP production. Similarly, 100 μM of DFP also severely inhibited glycolysis, the glycolytic reserve, and the glycolytic reserve capacity after 5 days treatment. The same trend was observed in T47D cells, where 100 μM of DFP inhibited OCR and ECAR after 5 days of treatment (Figure 6).

### 3.4. DFP Induces ROS and Mitochondrial Superoxide Production

One hypothesis for explaining the underlying metabolic effects of DFP is the induction of oxidative stress. To test this hypothesis, here we used CM-H2DCFDA to quantitatively measure the total cellular production of reactive oxygen species (ROS). Figure 7a,b showed that treatment with DFP (from 1 to 50 μM) for 5 days elevated ROS production by up to nearly three-fold. Next, to estimate the mitochondrial contribution to DFP-induced ROS production, we used the MitoSox Red probe to specifically measure the mitochondrial superoxide anion levels [32]. Figure 7c indicates that DFP treatment (from 1 to 50 μM) for 5 days clearly increased the superoxide production by ~two-fold. Therefore, the mitochondrial ROS production was a key component of the total cellular ROS production induced by the DFP treatment. In addition, we evaluated the total cellular ROS production and mitochondrial ROS production in the presence of N-acetyl-cysteine.

Figure 8a shows that treatment with DFP (1 μM to 10 μM) in the presence of NAC 1 mM did not increase the ROS production in the MCF7 cell line. Figure 8b,c shows that 1 mM of NAC was sufficient to revert the mitochondrial ROS production phenotype promoted by DFP treatment in the MCF7 and T47D cell lines. Therefore, we conclude that DFP treatment is sufficient to significantly increase mitochondrial ROS production, which could mechanistically explain some of its metabolic effects.

## 4. Discussion

In this report, we currently examined the repurposing of Deferiprone (DFP) for the targeting of mitochondrial organelles in CSCs. DFP (i.e., Ferriprox) is a clinically approved iron chelator which is normally used for the treatment of iron overload, such as in patients with thalassaemia major. Importantly, iron is a required co-factor for the proper functioning of mitochondria as metabolic organelles, such as in the production of ATP by OXPHOS. In this context, we employed MCF7 and T47D cells as a model system for ER(+) breast cancer, which is the most common sub-type of breast cancer, representing nearly 60–70% of all breast cancer cases worldwide.

Here, we used a 3D tumorsphere formation as an assay to monitor the propagation of CSCs in an anchorage-independent fashion. More specifically, we observed that DFP potently blocked 3D tumorsphere formation under low attachment conditions, revealing an IC-50 of ~100 nM for the MCF7 cell line and an IC-50 of ~0.5 to 1µM for the T47D cell line. Remarkably, DFP is thus one of the most potent clinically approved drugs that we have discovered. Interestingly, we now show that DFP at concentrations of 50µM and 100µM effectively targeted both the glycolytic and mitochondrial-based metabolism, resulting in significant reductions in critical metabolic parameters such as OCR and ECAR. Lastly, we showed that DFP treatment at lower concentrations resulted in elevated ROS production and mitochondrial superoxide production. As such, we now suggest that DFP may be a candidate for drug repurposing for the more effective targeting and eradication of CSCs. Interestingly, other clinically approved iron chelators are known and may have similar anti-CSC activity, such as Desferrioxamine (DFO, Desferal) and Deferasirox (Exjade or ICL-670) [33,34,35].

Therefore, Phase II clinical trials of the various iron chelators may be warranted. One clear advantage of Deferasirox and Deferiprone (DFP) is that they are both orally active, allowing them to be used on an outpatient basis without intravenous administration [36,37]. However, Deferasirox showed the highest toxicity when evaluated using Zebrafish embryo mortality [38]. Deferiprone (DFP) was non-toxic up to a concentration of 500 µM in an in vivo model [37]. As a result, Deferiprone (DFP) may be clinically preferred as an anti-cancer treatment.

## 5. Conclusions

Here, we showed that DFP can effectively target CSCs in a dose-dependent manner while showing little or no effect on stromal (hTERT-BJ1 fibroblasts) and normal epithelial cells (MCF10A). In fact, in both our breast cancer models (MCF7 and T47D) low concentrations of DFP were sufficient to promote cellular and mitochondrial ROS production. The oxygen consumption rate and extra cellular acidification rate were both significantly affected by cellular and mitochondrial ROS production. For this reason, CSCs that are more mitochondrial-dependent were more drastically affected by DFP treatment at significantly lower concentrations as compared with adherent cell monolayers. Moreover, these results suggest that an increase in the mitochondrial ROS production by iron chelation is a new key mechanism to prevent CSC propagation.

## Figures and Tables

**Figure 1 cells-09-01529-f001:**
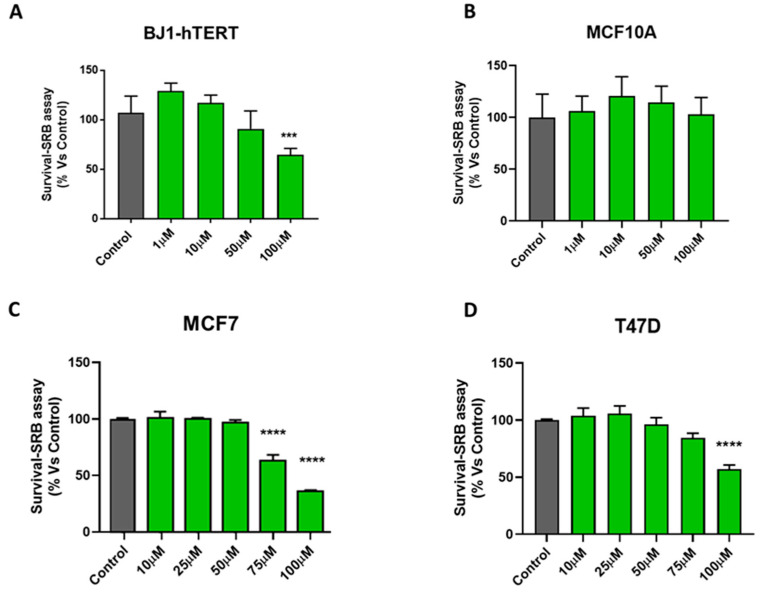
Effects of deferiprone (DFP) on cell viability in MCF7, T47D, hTERT-BJ1, and MCF10A cells. To evaluate the effects of DFP on cell viability, we used the sulphorhodamine (SRB) assay in hTERT-BJ1 fibroblasts, MCF10A, MCF7, and T47D breast cancer cells. (**A**,**B**) Note that ~70% of hTERT-BJ1 fibroblasts and nearly 100% of MCF10A remained viable at 100 μM of DFP treatment after 5 days of treatment. (**C**,**D**) In contrast, DFP dose dependently inhibited cell viability in MCF7 and T47D cell monolayers after 5 days of treatment, with an IC-50 of between 75 and 100 μM. *** *p* < 0.0001; **** *p* < 0.00001.

**Figure 2 cells-09-01529-f002:**
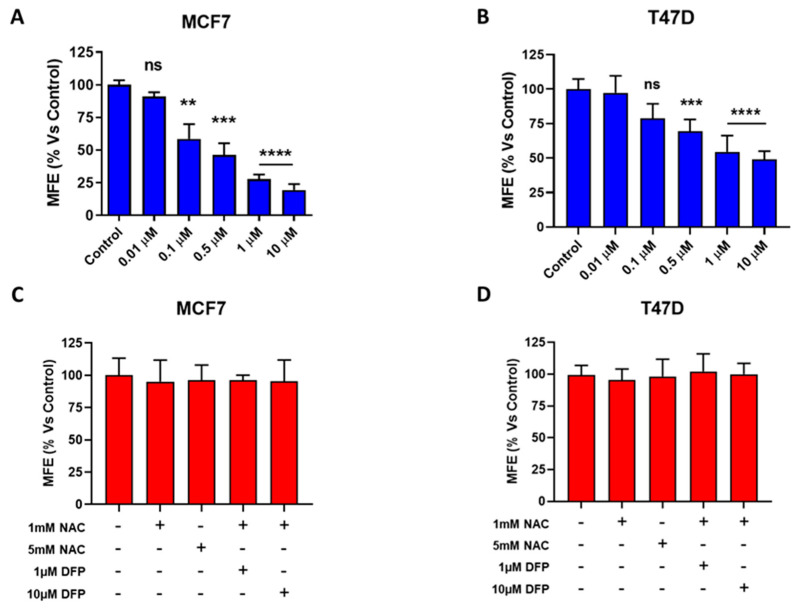
DFP inhibits cancer stem cell (CSC) propagation in MCF7 and T47D cells. We used a 3D tumorsphere assay to as a read-out to measure the CSC activity. This assay quantitates the functional ability of CSCs to undergo anchorage-independent growth under low-attachment conditions. MFE = Mammosphere Formation Efficiency. (**A**) Note that DFP potently inhibits 3D anchorage-independent growth, with an IC-50 of ~100 nM, after 5 days of treatment. ns = not significant; ** *p* < 0.001; *** *p* < 0.0001; **** *p* < 0.00001. (**B**) Note that DFP potently inhibits 3D anchorage-independent growth, with an IC-50 of ~0.5 to 1 μM after 5 days of treatment. ns = not significant; *** *p* < 0.0001; **** *p* < 0.00001. (**C**,**D**) DFP treatment at 1 and 10 μM in the presence of 1 mM and 5 mM of N-acetyl-cysteine (NAC).

**Figure 3 cells-09-01529-f003:**
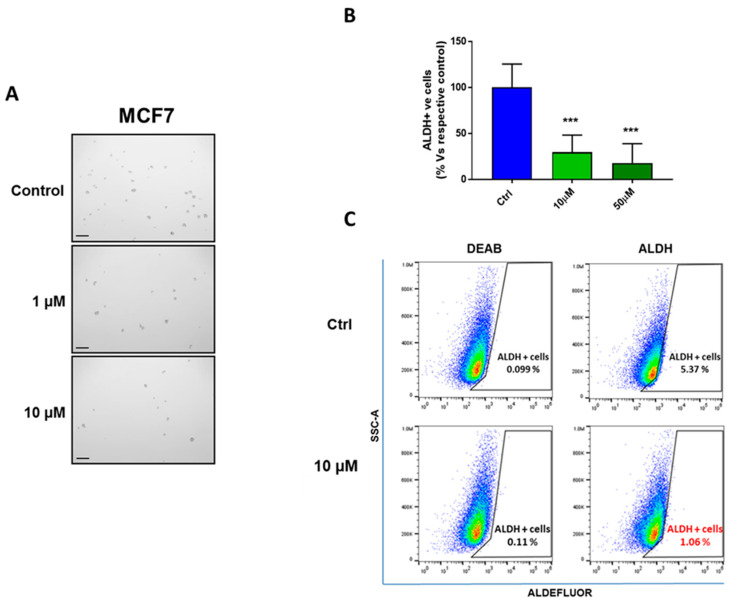
DFP reduces the ALDH activity in MCF7 cells. (**A**) Representative images of CSCs in anchorage-independent growth (control vehicle alone, 1 μM, and 10 μM are shown, respectively). A scale bar of 500 μM is shown. (**B**) ALDH is another established marker of “stemness” and the Epithelial-Mesenchymal Transition (EMT) in cancer cells. Therefore, we used the ALDH activity to further validate the effects DFP on CSCs. Note that 50 μM of DFP reduced the ALDH activity by >75% after 5 days of treatment. *** *p* < 0.0001. (**C**) Representative images of ALDEFLUOR assays performed by FACS (Sony SH800) and analyzed using the FlowJo Software (BD) (control vehicle alone and 10 μM are shown, respectively).

**Figure 4 cells-09-01529-f004:**
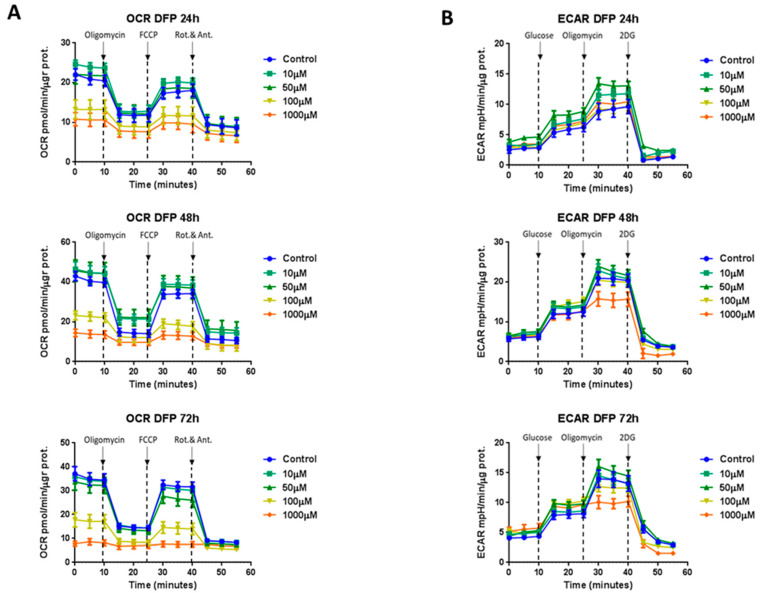
DFP reduces the mitochondrial and glycolytic metabolism in MCF7 cells: 24 to 72 h of treatment. (Panel **A**) Note that the DFP treatment significantly inhibited the oxygen consumption (OCR) in a time-dependent and dose-dependent fashion, showing a consistent reduction in oxygen consumption already at a concentration of 100 μM. Here are shown the 24, 48, and 72 h time points, respectively. (Panel **B**) DFP treatment slightly reduced the extracellular acidification after 72 h treatment at a concentration of 1 mM. Here are shown 24, 48, and 72 h time points, respectively.

**Figure 5 cells-09-01529-f005:**
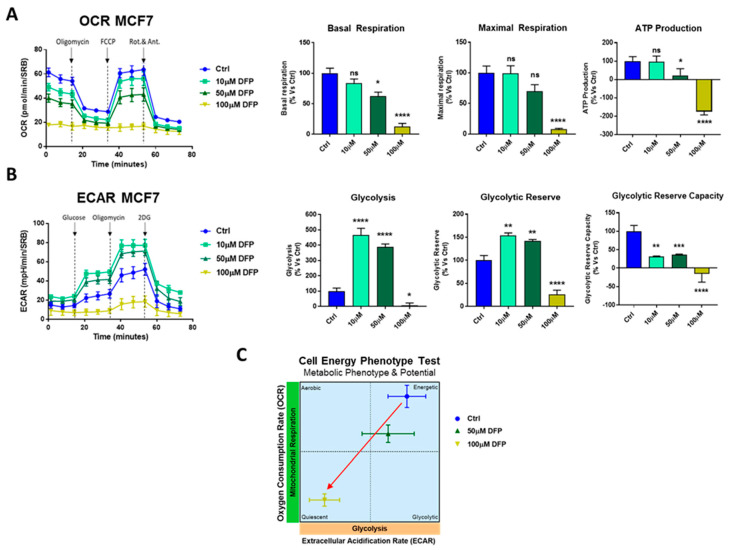
DFP reduces the OCR and extracellular acidification rates (ECAR) in MCF7 cells: 120 h of treatment (5 days). (**A**) Note that DFP dose dependently inhibited OCR, with maximal effects at 100 μM, resulting in severe ATP depletion. Significantly, basal respiration reduction and ATP depletion were also observed at 50 μM. ns = not significant; * *p* < 0.01; ** *p* < 0.001; **** *p* <0.00001. (**B**) Note that DFP dose dependently inhibited ECAR, with maximal effects at 100 μM, resulting in the near complete inhibition of glycolysis. ns = not significant; * *p* <0.01; ** *p* < 0.001; *** *p* < 0.0001; **** *p* < 0.00001. (**C**) OCR vs ECAR metabolic phenotype and potential is represented by following the manufacturer’s guidelines and instructions (Seahorse, Agilent Technologies). The energy plot shows that the 100 μM DFP-treated cells are clearly metabolic quiescent compared with the vehicle alone-treated cells.

**Figure 6 cells-09-01529-f006:**
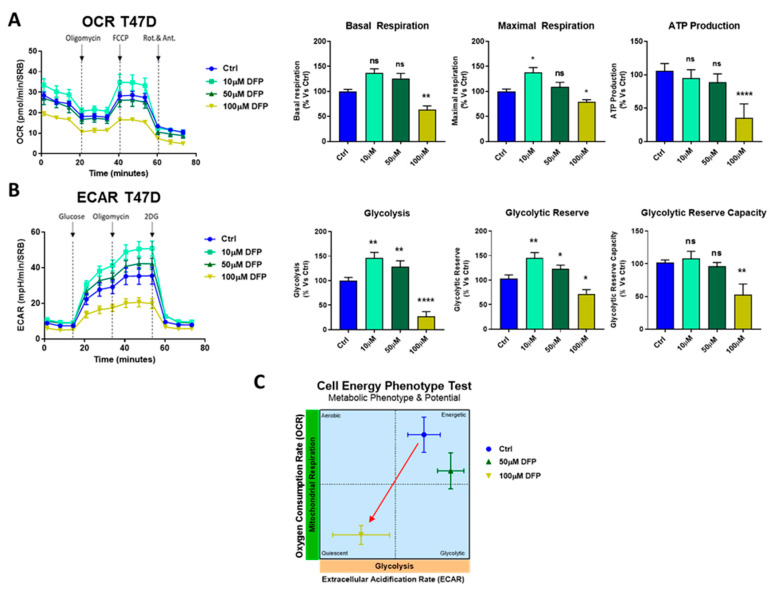
DFP reduces OCR and ECAR in T47D cells: 120 h of treatment (5 days). (**A**) Note that DFP dose dependently inhibited OCR, with maximal effects at 100 μM, resulting in severe ATP depletion. Significantly, basal respiration reduction and ATP depletion were also observed at 100 μM. ns = not significant; * *p* < 0.01; ** *p < 0.001*; **** *p* <0.00001. (**B**) Note that DFP inhibited ECAR, with maximal effects at 100 μM, resulting in the near complete inhibition of glycolysis. ns = not significant; * *p* < 0.01; ***p* < 0.001; **** *p* < 0.00001. (**C**) OCR vs ECAR metabolic phenotype and potential is represented by following the manufacturer’s guidelines and instructions (Seahorse, Agilent Technologies). The energy plot shows that the 100 μM DFP-treated cells are clearly metabolic quiescent compared with the vehicle alone-treated cells, as we showed in MCF7 cells.

**Figure 7 cells-09-01529-f007:**
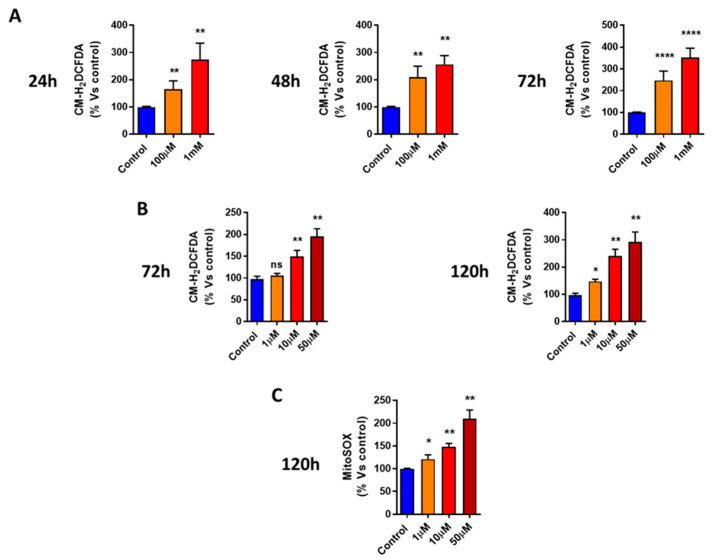
DFP induces total cellular reactive oxygen species (ROS) production in MCF7 cells. (**A**) Note that DFP increases the total cellular ROS production in a time-dependent and concentration-dependent manner (from 24 h to 72 h), with maximal effects observed at 1 mM. ** *p* < 0.001; **** *p* < 0.00001. (**B**) DFP treatments at 72 h and 120 h significantly elevate the total cellular ROS production after DFP treatment in the micro-molar range (1 to 50 μM). ns = not significant; * *p* < 0.01; ** *p* < 0.001. (**C**) DFP induces mitochondrial superoxide production in MCF7 cells and significantly elevates mitochondrial superoxide levels after DFP treatment in the micro-molar range (1 to 50 μM). ns = not significant; * *p* < 0.01; ** *p* < 0.001.

**Figure 8 cells-09-01529-f008:**
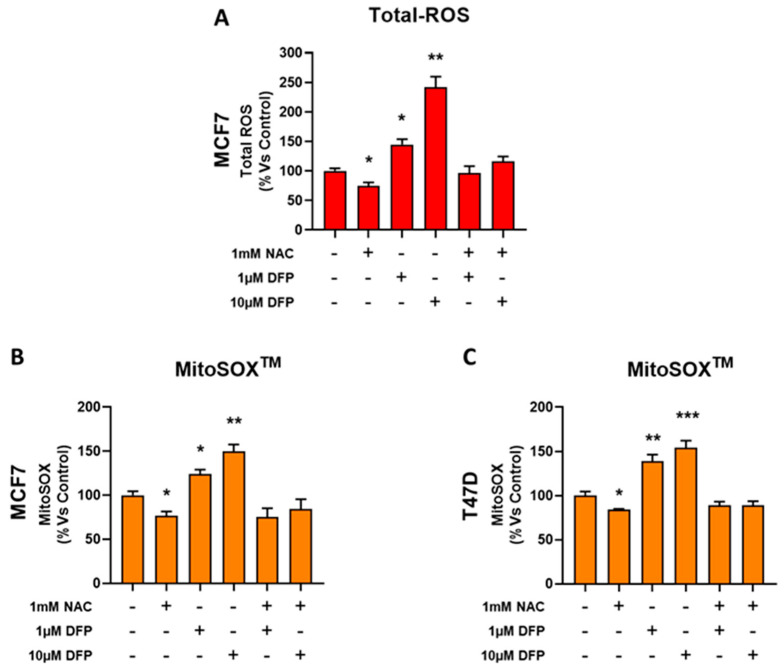
NAC reverted the DFP effect by restoring REDOX balance. (**A**) Note that DFP increases the total cellular ROS production in a concentration-dependent manner (from 1 μM to 10 μM). In the presence of 1 mM of NAC, the effect of DFP was reverted and the total cellular ROS content was similar to the control (vehicle alone). * *p* < 0.01; ** *p < 0.001*. (**B**,**C**) Similar trends were obtained by measuring the mitocondrial ROS content in MCF7 and T47D cells. Here, again the effect of DFP was reverted by NAC at concentrations of 1 mM and 10 mM. * *p* <0.01; ** *p* < 0.001; *** *p* < 0.0001.

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
