# Peer review of "Deferiprone (DFP) Targets Cancer Stem Cell (CSC) Propagation by Inhibiting Mitochondrial Metabolism and Inducing ROS Production"

_cells, 2020, doi:10.3390/cells9061529_

Round 1

Reviewer 1 Report

The work reported by Fiorillo et al is a nice example for drug repurposing. However, the points below need to be considered:

  • Additional cell lines should be screened other than MCF-7
  • Figures have low quality especially Fig. 1
  • There is a typo in legend of Fig. 1 and 2 where the unit of the concentration is not properly written

Author Response

As suggested by the reviewer, we screened more cell lines: the T47D cell line and a non-tumorigenic epithelial cell line, namely MCF10A, have both been added, for comparison purposes.

As suggested by the reviewer, we have now improved the quality of Figure 1.

As suggested by the reviewer, we revised and corrected the legends of Figures 1 and 2, and we corrected the units of concentration.

Reviewer 2 Report

This is a very interesting manuscript revealing the effect of the iron chelator deferiprone on cancer stem cell (CSC). It is very well written and is of interest for researchers working in oncology.

However, there are some points that should be carefully addressed in order to improve the manuscript. The most important question is related with the mechanistic support of the main finding of the article. Authors reported an effect of DFP on mitochondrial metabolism and ROS production when the drug is used in the range of micromolar. However, its effect on CSC is shown in the range of nanomolar. Therefore, the mechanism involved in the inhibition of CSC propagation is not related with mitochondrial metabolism and ROS production as the manuscript suggests. This question should be revised. Regarding this issue, concentrations range reported in lines 67 and 69 should be checked.

In addition, discussion should be elaborated more in depth, including dissertation about potential mechanisms and comparison with other studies.  Text in the “Conclusions” section should be moved to “Discussion”. A more strict conclusion should be indicated.

Other minor points:

-Lines 156-158: A citation is necessary.

-Line 163: Please, indicate what EMT stands for.

-Line 210: “celluar”

Author Response

As suggested by the reviewer, we now more carefully highlight the effects of DFP on ROS production that can specifically affect CSC propagation, especially at low concentrations.

The increased production of ROS based on iron depletion only slightly affects the mitochondria during the time window at low concentrations, but it is clearly enough to reduce the ability of the cells to divide and generate CSCs.

As suggested by the reviewer, we have corrected the concentrations ranges, in lines 67 and 69 of the text.

As suggested by the reviewer, we have now better elaborated our conclusions and we have moved the previous text from the Conclusion sections to Discussion section.

Moreover, we add citations in lines 156-158 (now lines 168-169), we have now specified Epithelial-Mesenchymal Transition (EMT) and we corrected the typographical error in line 210 (now line 245).

Reviewer 3 Report

In this manuscript Fiorillo et al. analyzed the effect of the iron chelator deferiprone (DFP) on tumorsphere formation and growth in monolayer of MCF-7 breast cancer cells. They show that it inhibits tumorsphere formation with an IC50 of approximately 100 nM, suggesting that it might inhibit propagation of cancer stem cells. In monolayer, the IC50 is much higher indicating a more robust effect and specificity on cancer stem cells. Finally, they detect defects in mitochondrial respiration and glycolysis, and increased ROS levels after treatment of MCF7 monolayers with DFP.

Major points

  1. The authors used fibroblasts BJ1-hTERT as control to test toxicity of DFP in normal cells (Figure 1A). However, they are using MCF7 cells, i.e. epithelial cells, as experimental model. MCF-10A cells, a well-known model for normal human mammary epithelial cells, should instead be used.
  2. Another breast cancer cell line, ER+ and PR+ like MCF7, should be used to confirm the results obtained with MCF7, at least in the key experiments, to strengthen the conclusions. T47D cells are an example.
  3. Figure 4: the authors show the effects of DFP on oxygen consumption and glycolysis 120 days after treatment. Is there any effect at early time points? Disruption of the energy metabolism may be a consequence and not the cause of the phenotype.
  4. Figure 5C: production of mitochondrial superoxide should be tested at earlier time points (not only 120 h after treatment).
  5. Figure 1: ALDH activity should also be measured in tumorspheres.
  6. If increased ROS production is the key mechanism involved in DFP-mediated inhibition of tumorsphere formation, co-treatment with anti-oxidants should restore tumorsphere formation. The authors should perform this experiment to substantiate the mechanism they propose.

Minor points

  1. Lines 55-56: the authors assess that many iron chelators have been shown to inhibit tumor growth but they only cite two references.
  2. Discussion: it’s very synthetic, a summary of the results. The authors could elaborate more on the significance of their results, comparing the effects of DFP with other iron chelators in this section instead of discussing them in the Conclusions.
  3. There are some typo errors. For example, line 32: “under” should read “undergo”.

        Figure 4B: “phenothype” should read “phenotype”. Line 225. Line 233. In some figure legends micro (u)M is missing.

Author Response

As suggested by the reviewer, we have now used the MCF10A cell line (normal human mammary epithelial cells), as an additional control to assess DFP toxicity.

As suggested by the reviewer, we have used T47D cells as an additional cell line, to reproduce the key findings from MCF7 related experiments.

In Figure 4 (now figure 5), we now show OCR and ECAR profiles after 120 hours of DFP treatment.

In the previous Figure (now figure 4), we represented the effects of DFP at a different time point. We agree with the reviewer that disruption of energy metabolism may be a consequence and not the cause of the phenotype and we highlight that in the Discussion and Conclusion sections.

As suggested by the reviewer, we have now used N-acetyl-cysteine (NAC) as an anti-oxidant. Co-treatment with DFP 1µM and 10 µM NAC restored the mammosphere formation (see Figure 2, panels C-D).

As suggested by the reviewer, we better elaborate the Discussion and Conclusion sections.

As suggested by the reviewer, we have corrected the typographical errors in line 32 (now line 33).

As suggested by the reviewer, we have corrected the typographical error in figure 4B (now figure 5C) and we have corrected the typographical error in line 225 and 233 (now line 263 and 284).

Round 2

Reviewer 2 Report

Authors have adequately attended reviewers´ suggestions.

Reviewer 3 Report

The authors performed new experiments that strengthen the conclusions. The manuscript can be accepted for publication.